# Cervical Cytology–Histology Correlation Based on the American Society of Cytopathology Guideline (2017) at the Russian National Medical Research Center for Obstetrics, Gynecology, and Perinatology

**DOI:** 10.3390/diagnostics12010210

**Published:** 2022-01-15

**Authors:** Aleksandra Asaturova, Darya Dobrovolskaya, Alina Magnaeva, Anna Tregubova, Guldana Bayramova, Gennady Sukhikh

**Affiliations:** FSBI “National Medical Research Centre for Obstetrics, Gynecology and Perinatology Named after Academician V.I.Kulakov” of the Ministry of Health of the Russian Federation, 4, Oparina Street, 117513 Moscow, Russia; dashaGRI@yandex.ru (D.D.); alinamagnaeva03@gmail.com (A.M.); a_tregubova@oparina4.ru (A.T.); bayramova@mail.ru (G.B.); sukhikh@oparina4.ru (G.S.)

**Keywords:** cervical cancer, cytology–histology correlation, ASC guideline, discrepancy analysis

## Abstract

Recent evidence suggests that a cytology–histology correlation (CHC) with discrepancy detection can both evaluate errors and improve the sensitivity and specificity of the cytologic method. We aimed to analyze the errors in cytologic–histologic discrepancies according to the CHC protocol guideline of the American Society of Cytopathology (2017). This retrospective study included 273 patients seen at the National Medical Research Center of Obstetrics, Gynecology and Perinatology (Moscow, Russia) between January 2019 and September 2021. The patients’ mean age was 34 ± 8.1 years. The cytology–histology agreement was noted in 158 cases (57.9%). Major discrepancies were found in 21 cases (7.6%), while minor discrepancies were noted in 93 cases (34.1%). The reason for 13 (4.8%) discrepancies was a colposcopy sampling error and, in 46 (16.8%) cases, the reason was a Papanicolaou (PAP) test sampling error. The discrepancy between primary and reviewed cytology was due interpretive errors in 13 (4.8%) cases and screening errors in 42 (15.4%) cases. We demonstrated that the ASC guidelines facilitate cervical CHC. A uniform application of these guidelines would standardize cervical CHCs internationally, provide a scope for the inter-laboratory comparison of data, and enhance self-learning and peer learning.

## 1. Introduction

Cervical cancer is an important health and socioeconomic issue worldwide [1,2,3]. The recent decline in cervical cancer incidence and mortality is associated with the growing availability of cytological screening. Despite the fact that the high-risk human papillomavirus (hrHPV) test is considered an alternative to primary cervical cancer screening [4], cytology smear remains the most successful of the cervical cancer prevention programs developed to date, especially in many low-income countries [5]. Obviously, the simultaneous implementation of a nationwide HPV vaccination and an effective cervical cancer screening program can significantly reduce mortality from cervical cancer.

In Russia, cervical cancer ranks number six in incidence and number ten in mortality among women. The mean patient age is 52.6 years, with the majority of cases occurring between 40 and 44 and between 55 and 59 years of age [6]. These findings agree with the observations in countries with similar economic conditions. In Russia, a public funds-based model of cervical cancer screening has been in place for several years. According to the order of the Ministry of Health of the Russian Federation, the Papanicolaou (PAP) test is used as a screening method [7].

The PAP test is a complex, moderately sensitive and highly specific laboratory method whose sensitivity and specificity depend on the adequacy of samples, highly qualified laboratory processing, and professional interpretation [8]. To analyze the causes of both false-negative (FN) and false-positive (FP) results, a cytologic–histologic correlation (CHC) can be applied [9,10,11].

CHC is a special method used to determine whether the results are concordant or discordant and to evaluate errors in cytologic screening [12]. It can be a powerful quality assurance tool and feedback mechanism for specialists. Literature reviews have reported that CHC discrepancy accounts for approximately 11–28% of all cytology-biopsy cases as a result of interpretive, screening, and sampling errors [13,14,15].

Before the protocol of CHC was established, laboratories in many countries tried to develop their own discrepancy analysis algorithms, but since each algorithm was different, comparing the quality of these laboratory diagnostics, analyzing the data, and making the right treatment decisions were difficult [16]. The first studies evaluating CHC recorded discrepancies only for benign and atypical squamous cells of undetermined significance (ASC-US) without taking into account the stage of the lesion [11,15,17,18,19].

To eliminate these limitations and to come to an agreement, the College of American Pathologists Gynecologic Cytopathology Quality Consensus Conference working group 4 and the Clinical Practice Committee of the American Society of Cytopathology (ASC) formulated guidelines to provide a discrepancy assessment grid of the CHC protocol [9,20]. The protocol includes intervals between cytologic and histologic specimens, data correlations, search logistics, correlation definitions, and calculated parameters [20]. Taking into account all of the above, the method of cytologic–histologic correlation is highly valuable for improving diagnostic testing and screening processes and is crucial for standardizing CHC protocols [20,21].

We aimed to analyze the errors causing cytologic–histologic discrepancies according to the CHC protocol guidelines of the ASC (2017). CHC was used as a quality assurance exercise at the National Medical Research Center of Obstetrics, Gynecology and Perinatology (Moscow, Russia) among patients undergoing opportunistic screening of the cervix with a two-step difference between cytology and corresponding biopsy for classifying discrepancies.

## 2. Materials and Methods

This study was a retrospective cross-sectional descriptive study consisting of 327 consenting women aged 20 to 70 who had undergone an opportunistic screening for cervical cancer that included liquid-based cervical cytology (LBC), conventional cervical smears, and a histologic examination at the National Medical Research Center of Obstetrics, Gynecology and Perinatology (Moscow, Russia) between January 2019 and September 2021.

The histologic material was obtained from cervical colposcopy-directed biopsy (CDB), endocervical curettage, loop electrosurgical excision (LEEP), cone biopsies, and hysterectomies and was paired with corresponding PAP tests. The inclusion criteria included up to a 6-month interval between cervical smear and biopsy. The exclusion criteria were the following: (1) insufficient data, (2) pregnancy, and (3) a period between cytologic and histologic tests > 6 months. 

Cytology and histology reviews were conducted in accordance with The Bethesda System (TBS) (2014) [8] and the World Health Organization (WHO) classification, respectively [22]. The colposcopic data were interpreted with the International Federation for Cervical Pathology and Colposcopy (IFCPC) nomenclature [23].

Colposcopy was performed for all reports with abnormal cytology (ASC-US or more severe lesions, and glandular lesions with atypical glandular cells, not otherwise specified (AGC-NOS) or more severe lesions) and persistence of hrHPV without cytology abnormal cells following the Russian guidelines [24,25]. For positive tests with acetic acid (VIA), concurrent cervical cytology and colposcopy-directed biopsy were provided to minimize loss to follow-up for these patients (since many of them were referred to us from different regions of Russia and countries of the Commonwealth of Independent States (CIS)). HPV testing was carried out with PCR real-time (DNA-Technology LLC, Russia).

Cervical smears were screened initially by general cytopathologists. An analysis of the quality assurance measure was performed using the published guidelines of the ASC (2017) for CHC, and CHC was assessed individually for each patient. According to the CHC protocol, a two-step difference between the cervical cytology result and the corresponding biopsy was considered a major discrepancy, while a one-step difference was considered a minor discrepancy. Pairs with an exact agreement were designated as an agreement [20].

A major undercall is defined as a negative cytology result for intraepithelial lesions or malignancy (NILM), while a biopsy shows high-grade squamous intraepithelial lesion (HSIL) or an adenocarcinoma. In addition, cytological result ASC-US/AGC-NOS and a corresponding biopsy with squamous CA should be also considered as a major undercall.

A minor overcall/undercall was attributed to the one-step difference between cytology and histology. Minor variance was attributed to a discrepancy not suitable to be categorized as under- or overcall.

The cytology–histology pairs of all discrepant cases (major or minor) were reviewed independently by gynecological cytopathologists. After this review, a more experienced gynecologic cytology cytopathologist made corrections and gave recommendations to the general cytopathologist who initially signed out the case.

A true-positive (TP) cytology review was defined as a positive cytology review with a corresponding positive histology review. A true-negative (TN) cytology review was defined as a negative cytology review with a corresponding negative histology review. A false-negative (FN) cytology review was defined as a negative cytology review with a corresponding positive histology review. A false-positive (FP) cytology review was defined as a positive cytology review with a corresponding negative histology review.

The positive predictive value (PPV) of the cytology review was defined as the ability of cervical cytology to predict abnormalities in the following histologic review.

The statistical analysis was performed using IBM SPSS Statistics 26. For the demographics and baseline characteristics, descriptive statistics were reported. For categorical variables, the number and percentage of participants in each category were calculated; for continuous variables, the total number of participants, median, first quartile (Q1), second quartile (Q2), and third quartile (Q3) were calculated. The number of false-negative and false-positive cytology cases was compared with the number of all cases, and thus false-negative and false-positive rates were obtained. The following formulae were used [6]:

Specificity was estimated using the following formula:(1)$$\frac{No.\;\;of\;TN}{No.\;\;of\;TN+No.\;\;of\;FP}\times 100$$

Sensitivity was estimated using the following formula:(2)$$\frac{No.\;\;of\;TP}{No.\;\;of\;TP+No.\;\;of\;FN}\times 100$$

Positive predictive value (PPV) was calculated using the following formula:(3)$$\frac{No.\;\;of\;True\;Positives\;Cytology\;Diagnosis}{No.\;\;of\;True\;Positives\;Cytology\;Diagnosis+No.\;\;of\;False\;Positives\;Cytology\;Diagnosis}\times 100$$

The gold standard for calculation of sensitivity/specificity/PPV was gynecologic cytopathologists. This study was approved by the Ethics Committee of the National Medical Research Center of Obstetrics, Gynecology and Perinatology (Moscow, Russia) (Protocol No. 2, 11 March 2021).

## 3. Results

Between January 2019 and September 2021, 273 patients met the eligibility criteria and were selected after 327 patients were screened. The mean period between the cervical cytology and corresponding histology was 23 days (range: 0–105 days). The mean age was 34 ± 8.08 years (IQR: 30–40) (Table 1). Of all patients, 214 underwent colposcopy in accordance with the Russian guidelines (2020) [22]. CBD was performed simultaneously with cytology in 54 cases (19.8%). No colposcopically abnormal findings were found in 37 cases (13.6%), while 124 cases (45.4%) and 53 cases (19.4%) had minor and major colposcopic abnormal features, respectively.

HPV status was known in 187 cases (68.5%). The results of the cytology–histology discrepancies in hrHPV patients are presented in Table 2. No statistically significant correlation was found between HPV status and CH discrepancy (*p* = 0.221).

The results of the study are shown in Table 3. Cytologic–histologic agreement was reached in 159 cases (58.2%), where the majority of cases (33.7%) showed high-grade squamous intraepithelial lesions (HSILs) or atypical squamous cells, cannot exclude HSIL (ASC-H).

Major discrepancies were found in 21 cases (7.7%). In 13 cytologic reports (4.8%) of NILM classified as major undercall, the histologic report showed HSIL or carcinoma in situ and above (CIS+). Eight cases (2.9%) were interpreted as major overcalls: cytologic smears reported HSIL, while histology did not reveal atypical cells.

By contrast, minor discrepancies were found in 88 cases (34.2%). Of the 67 cases (24.5%) classified as minor undercall, 24 cases (8.8%) showed NILM on cytology with a biopsy report of LSIL, 6 cases (2.2%) showed ASC-US upon cytology with a biopsy report of HSIL, 13 cytologic smears (4.8%) of the LSIL histologic review showed HSIL, 2 cases (0.7%) showed LSIL upon cytology with a histologic report of carcinoma in situ and above (CIS+), and 22 cytologic smears (8.1%) of the HSIL biopsy displayed features of CIS+. Minor overcall was detected in 21 cases (7.7%). In 14 cytologic reports (5.1%) of LSIL, the histologic review did not reveal atypical cells. Seven cases (2.6%) showed HSIL upon cytology with a biopsy report of LSIL.

Minor variances were seen in three cases (1.1%) with ASC-US upon cytology with a histologic report without atypical cells and in two cases (0.7%) with adenocarcinoma in situ and above (AIS+) upon cytology with a histologic diagnosis of HSIL.

According to the ASC guidelines, CHC was performed on all 114 discrepancy cases (41.8%) (Table 4). Then, these cases were reviewed by gynecological cytopathologists, so cytologic–histologic agreement was reached in 33 cases (12.1%), while 81 cases (29.7%) still had discrepancies. The minor variance was noted in five cases (1.8%). Major discrepancies were found in 18 cases (6.6%), of which 13 cases (7.8%) and 5 cases (1.8%) were classified as undercall and overcall, respectively. At the same time, minor discrepancies were recorded in 58 cases (21.2%), of which 49 cases (17.9%) and 9 cases (3.3%) were categorized as minor undercall and overcall, respectively.

The CHC and error results obtained are presented in Figure 1. The majority of discrepancies were associated with PAP test sampling errors (16.8%), while the least common errors were interpretative errors (4.8%).

An exact agreement between HSIL (PAP test) and HSIL (histology) was established in 102 cases (37.4%), including 92 cases based on general cytopathologist diagnoses plus 10 cases based on gynecological cytopathologist diagnoses. The major discrepancy between the HSIL (PAP test) and FP results was 4.4%. TP results were found in 163 cases (59.7%), and TN results were found in 48 cases (17.6%). Additionally, FN results were found in 37 cases (13.6%), and FP results were found in 25 cases (9.2%). The sensitivity and specificity of the cytologic method were 87.6% and 64%, respectively. A Positive Predictive Value (PPV) accounted for 74.6% of cases.

## 4. Discussion

In this study, we investigated considerable errors in cytologic–histologic discrepancies according to the guideline of the American Society of Cytopathology. We applied this guideline for the first time at the National Medical Research Center of Obstetrics, Gynecology and Perinatology (Moscow, Russia). The main purpose of this study was to identify the causes of cytology–histology discrepancies to further improve the cytological diagnostic quality. We found that PAP test sampling errors were the most frequent causes of discordant results. The percentage of discrepancy based on the abovementioned guideline is consistent with those of other studies and suggests that CHC protocol application could allow us to both classify discordant cases and identify their causes.

We plan to use the CHC protocol to standardize laboratory work and assess the sensitivity and specificity of the cytological method in the local department as well as collaborative and sponsored laboratories.

Major discrepancy: The present study was designed to determine the major discrepancies between cytologic and histologic reviews. While the rate of major discrepancies was up to 7.7% in a general cytopathologist review, the application of the second review reduced the rate of major discrepancies to 6.6%. An underdiagnosis might be related to the misinterpretation of atypical metaplasia/atypical atrophy and reparative changes as HSIL, while the presence of atypical cells in atrophic smears and a small number of these cells could disrupt the identification of high-grade lesions on cervical smears. This rate is in agreement with the findings by Gupta (2019), which showed a major discrepancy of 7.1% [26]. Unfortunately, in a recent Korean study by Ouh et al. [27], the ASC protocol was not used. To compare these studies, we adapted the main parameters and obtained the percentage of major discrepancies (6.6%). Consequently, the present findings seem to be consistent with those of other studies.

HSIL: Our study found that 33.7% of routine cytology reviews showed almost exact agreement between HSIL (cytology) and HSIL (histology). In the other cases, a discrepancy of HSIL was found. The minor and the major discrepancies of HSIL (PAP test) after a gynecologic cytopathologist review were 1.38-fold lower and 1.6-fold lower than after primary cytology reviews, respectively. In addition, we demonstrated that the percentage of major discrepancies decreased after the revision from 7.7% to 6.6% in our study and from 6.4% to 4.4% in an Indian study [26]. Moreover, we demonstrated slightly more minor discrepancies (21.2% in comparison with 18.2% in the Indian research) and slightly less agreement (70.3% in comparison with 73% in the Indian research). Overall, these results indicate the need for an independent review of the cytology smears to reduce the range of false results and to improve laboratory competence.

The data regarding cervical pathology cytohistological correlation from Eastern European countries is only partly available. Nevertheless, the data that we have been able to investigate was similar to that of Russia based on economic and social background. Cervical cytology-based screening is the most widespread method in Belarus, Ukraine, Kazakhstan, Latvia, Lithuania and Serbia, although the HPV-based approach has also been approved for some categories of women [27,28,29,30,31,32]. The available data about the correlation between cytology and histology were published by Serbian researchers. For the cytology method, they reported a sensitivity of 87.3%, a specificity of 96.86%, a positive predictive value of 63.95%, total agreement in 41.09%, minor discrepancies in 45.54%, major discrepancies in 12.87%, and minor variance in 0.50%. Of note, both minor and major discrepancies were detected in many more cases in the Serbian cohort than in our cohort [33].

Time interval: The optimal time interval between cervical smear and biopsy is recommended at less than 6 months to avoid lesion regression [20,34]. We tried to reduce the loss to follow-up, so this period was minimized (a median of 23 days) in comparison to the study by Gupta et al. [26], where the median time to biopsy was 40 days.

hrHPV: Several recent studies investigated the impact of hrHPV on CHC results [35,36,37]. For example, a multicentric Korean study (Ouh et al.) found a higher risk for initial overcall in patients with hrHPV [35]. By contrast, our obtained findings did not show a strong relationship between HPV status and cytologic–histologic discrepancies (*p* = 0.221). A possible explanation for this contradiction between the current study and the previous Korean research may be the fact that we included not only women with known HPV statuses. Additionally, the results of our study indicate that a hrHPV-positive status was determined more frequently in women with HSIL (55.4%), while 23% of cytology smears in hrHPV-positive patients revealed NILM.

Colposcopy: Contrary to expectations, in two out of three of all cases with abnormal colposcopic findings (45.4% and 19.4% with minor and major abnormal colposcopic features, respectively), no histological abnormalities were observed [26,38,39].

Error analysis: Some previous papers reported a sampling discrepancy, a screening discrepancy or an interpretive discrepancy. Unfortunately, the approach focuses only on expected individual failures [33,40]. In this study, we tried to analyze all of these errors. In our series, the majority of the overcalls (major or minor) could be attributed to colposcopy sampling errors, while one case was attributed to an interpretative error. This finding is in agreement with, for example, those of Dodd (1993) and Jones (1996) [11,41]. The overall rate of cytologic sensitivity was 87.6%, with a specificity of 64% and a positive predictive value of 74.6%. Our study met the expected cytologic PPV and confirms the results of previous studies [17,42,43,44].

Particular attention should be paid to screening errors analysis. The cytologic–histologic agreement was reached in 33 cases after a gynecological cytopathologists’ review. Among these agreements there were NILM (n = 5), ASC-US (n = 6), ASC-H (n = 1) LSIL (n = 13) and HSIL (n = 8). When gynecological cytologists’ diagnoses were NILM, the general cytologists’ diagnoses were predominantly ASC-US, and it was revealed that reactive atypia could be mistaken for atypical cells with uncertain significance. When the gynecological cytologists’ diagnoses were ASC-US, the general cytologists’ diagnoses were predominantly HSIL because they considered the atypical cells as more altered and obviously malignant, while more experienced cytologists could not exclude the possibility that they could be reactive. In one case, when the gynecological cytologist’s diagnosis was ASC-US, a general cytopathologist diagnosed NILM due to inattention of whose the group of the atypical cells was missed. In one case, when a gynecological cytologist diagnosed ASC-H, a general cytologist diagnosed ASC-US because of the underestimation of the atypia degree. When the gynecological cytologists’ diagnoses were LSIL, the general cytologists’ diagnoses were predominantly NILM/ASC-US because koilocytos and squamous cells with mild dyskaryosis were misinterpreted for cells with degenerative changes. When the gynecological cytologists’ diagnoses were HSIL, the general cytologists’ diagnoses were predominantly ASC-US (n = 8) and LSIL (n = 4) due to underestimation of the atypia degree. In one case, when gynecological cytologists’ diagnoses were HSIL, a general cytologist diagnosed NILM due to inattention the group of the atypical cells was missed. Thus, we can highlight that attentive searching of the atypical cells groups and correct interpretation of the dyskaryosis degree could help to improve the general cytopathologists diagnoses. Consequently, open discussions that were considered in the process of revising provided additional knowledge for the general cytopathologists.

We strictly adhered to the guideline instructions and applied the CHC protocol only on discrepancy cases. By contrast, Gupta et al. (2020) analyzed all cytology–histology pairs. Although this study focused on cytohistologic correlations, the colposcopic findings were also analyzed in 78.4% of cases.

Some limitations to this study need to be acknowledged. First, the sample size was insufficient, so studies with a larger sample size are needed. This problem could be solved via interactions with other research centers while continuing our study. Second, the current research was not specifically designed to evaluate patients’ HPV status, so further data collection is required to determine the frequency of overcalls/undercalls in patients with hrHPV.

## 5. Conclusions

This project was undertaken to analyze errors in cytologic–histologic discrepancies according to the CHC protocol guidelines of the ASC (2017). The most obvious finding that emerged from this study is that the recent CHC protocol can be an applicable tool for specialists to determine whether results are concordant or discordant and to evaluate errors in cytologic screening. The results of this study indicate that the majority of discrepancies were associated with PAP test sampling errors at the National Medical Research Center of Obstetrics, Gynecology and Perinatology (Moscow, Russia). Notwithstanding the previously described limitations, this study suggests that future directions should be primarily related to the improvement of sampling techniques. Although the results corroborate the findings of a great deal of previous work in this field, further large-scale research in Russia is needed to estimate the correlation between colposcopic abnormal findings, hrHPV status, cytologic smears, and histologic results.

## Figures and Tables

**Figure 1 diagnostics-12-00210-f001:**
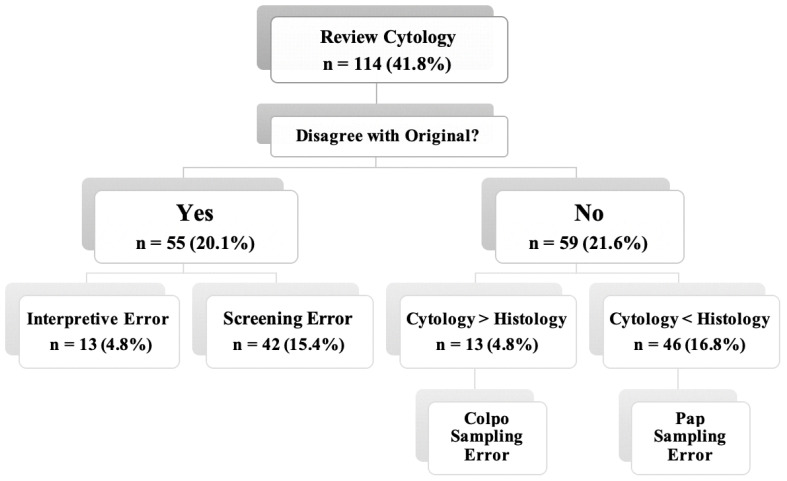
Evaluation of non-correlating cases according to the ASC guidelines [20]. **Interpretive Error**—abnormal cells were detected but incorrectly assessed by cytopathologists; **Screening Error**—abnormal cells were present in the smear but not detected upon screening; and **Sampling Error**—abnormal cells were absent in the cervical smear or biopsy.

**Table 1 diagnostics-12-00210-t001:** Comparison of mean age between cytologic, histologic, and colposcopic patient groups.

Cytology Diagnosis (BTS, 2014) (n = 273)	Age, Years		
	No (%)	Me	IQR (Q1–Q3)
NILM	86 (31.5)	31.5	27.0–39.0
ASC-US	10 (3.7)	35.0	31.0–46.0
LSIL	42 (15.4)	32.5	28.0–36.0
HSIL	104 (38.1)	35.0	30.0–41.0
ASC-H	13 (4.8)	41.0	31.0–38.0
AGS-NOS (endocervical)	14 (5.1)	38.0	30.0–48.0
CIS+	2 (0.7)	45.0	37.0–53.0
AIS+	2 (0.7)	31.0	30.0–32.0
**Histology Diagnosis** **(WHO, 2020)** **(n = 273)**			
Negative	74 (27.1)	34.0	30.0–42.0
LSIL (CIN1)	45 (16.5)	30.0	26.0–35.0
HSIL (CIN2–3)	128 (46.9)	34.5	30.0–38.5
>CIS	24 (8.8)	39.0	34.0–44.0
>AIS	2 (0.7)	32.0	26.0–38.0
**Colposcopic Score** **(IFCPC, 2011)** **(n = 202)**			
Normal	31 (15.3)	35.0	30.0–44.0
Minor colposcopic abnormal findings	117 (57.9)	32.0	29.0–35.0
Major colposcopic abnormal findings	54 (26. 8)	34.0	29.0–36.0

NILM, negative for intra-epithelial lesion or malignancy; ASC-US, atypical squamous cells of undetermined significance; LSIL, low-grade squamous intraepithelial lesion; ASC-H, atypical squamous cells, cannot exclude HSIL; HSIL, high-grade squamous intraepithelial lesion; AIS, adenocarcinoma in situ; CIS, carcinoma in situ.

**Table 2 diagnostics-12-00210-t002:** HPV status in discrepancy cases.

CHC	HPV		Total	*p*-Value
	Negative for hrHPV	hrHPV		
Agree, n (%)	29 (60.4%)	70 (50.4%)	99	0.221
Minor undercall, n (%)	10 (20.8%)	39 (28.1%)	49	
Major undercall, n (%)	2 (4.2%)	11 (7.9%)	13	
Minor variance, n (%)	2 (4.2%)	0	2	
Minor overcall, n (%)	4 (8.3%)	14 (10.1%)	18	
Major overcall, n (%)	1 (2.1%)	5 (3.6%)	6	
Total	48	139	187	

**Table 3 diagnostics-12-00210-t003:** Discrepancy assessment grid (over- and undercall refer to the cytology interpretation).

PAP Test N = 273	Biopsy Diagnosis Summary				
	Benign or Inflam	LSIL	HSIL	Squamous CA	>AIS
NILM, n (%)	49 (17.9%)	24 (8.8%)	12 (4.4%)	1 (0.4%)	0
ASC-US, n (%)	3 (1.1%)	1 (0.4%)	6 (2.2%)	0	0
LSIL, AGC-NOS, n (%)	14 (5.1%)	14 (5.1%)	13 (4.8%)	2 (0.7%)	0
HSIL, ASC-H, n (%)	8 (2.9%)	7 (2.6%)	92 (33.7%)	22 (8.1%)	0
>AIS, n (%)	0	0	2 (0.7%)	0	3 (1.1%)

Green—agreement; yellow—minor variance; orange—minor (undercall/overcall) discrepancy; and red—major discrepancy (undercall/overcall) cases. Percentage calculated for all included cases.

**Table 4 diagnostics-12-00210-t004:** Discrepancy assessment grid (over- and undercall refer to the cytology interpretation) revised by gynecologic cytopathologists.

PAP Test N = 114	Biopsy Diagnosis Summary				
	Benign or Inflam	LSIL	HSIL	Squamous CA	>AIS
NILM, n (%)	7 (2.6%)	18 (6.7%)	12 (4.4%)	1 (0.4%)	0
ASC-US, n (%)	5 (1.8%)	1 (0.4%)	1 (0.4%)	0	0
LSIL, AGC-NOS, n (%)	9 (3.3%)	13 (4.8%)	8 (2.9%)	1 (0.4%)	0
HSIL, ASC-H, n (%)	5 (1.8%)	0	10 (3.7%)	21 (7.7%)	0
>AIS, n (%)	0	0		0	2 (0.7%)

Green—agreement; yellow—minor variance; orange—minor (undercall/overcall) discrepancy; and red—major discrepancy (undercall/overcall) cases. Percentage calculated for all included cases.

## Data Availability

Data supporting reported results can be presented upon reasonable request.

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
