# Peer review of "Cervical Cytology–Histology Correlation Based on the American Society of Cytopathology Guideline (2017) at the Russian National Medical Research Center for Obstetrics, Gynecology, and Perinatology"

_diagnostics, 2022, doi:10.3390/diagnostics12010210_

Round 1

Reviewer 1 Report

Cervical cancer prevention programs are well developed now, however, in some parts of the world it is still not properly implemented. Moreover, the methods used or CC screening are different: Romanovsky, Pap test, HPV genotyping. The manuscript covers very important part of medicine – evaluation of the screening techniques. The study is well designed, enough materials were analyzed to perform reliable statistical analysis, the text in general is very well written. The references are up-to-date. However, some corrections should be made before acceptance and  publishing.

Comments

  1. A THOROW ENGLISH EDITING IS REQUIRED THROUGH THE TEXT.
  2. Title of the manuscript is too complex. Need to simplify and make it more understandable/clear for a potential audience. Which country national center? Is it really important to highlight that the data obtained from the natonal center? If yes, then need to specify which country.

Suggestions for the title: a. Cytology-histology correlation in cervical cytology based on the American Society of Cytopathology guidelines (2017): XXXX National Medical Research Center Results. b. Cervical Cytology vs. Histology correlation in cervical cytology based on the American Society of Cytopathology guidelines (2017): data from the national tertiary care center. Anyway, it is just a suggestion. The authors may create something better.

  1. Introduction - It is good to introduce potential readers into the details of the territory and population to ensure that an international audience will understand the proportion of analyzed materials. Please provide information about the female population in the region it age-distribution. It is important because cervical cancer has a bimodal age distribution, with the majority of cases occurring among women in their 30s and 40s, the age at which women are often raising families and ensuring the financial viability of their families and communities. Which method is used in the country for Cervical cancer screening?
  2. Line 37 in the introduction part – I suggest to be more careful with the definitions. Cytology smear is not = PAP test. Cervical cytology in the past was done by Romanovsky method, while PAP test abbreviation is related to the Papanicolaou method.
  3. The methods section does not include the detailed description of the statistical methods employed. Please make it more detailed as any original research theoretically should be reproducible.
  4. Inclusion/exclusion criteria need to be specified.
  5. The results part is described in a great details. Clear and interesting.
  6. The discussion part should be recomposed. The authors performed a very interesting study that should be discussed appropriately. Please cover the following

1.1       Rationale of the study (why it was done)

1.1.1    Main findings of the study

1.1.2    What makes our study unique

1.1.3    What it adds to what we already know

1.2       Study subjects

1.3       Subject of the discussion

1.4 Comparison of your results with neighboring countries studies results (if available), with countries of the same income levels  (may be some post-Soviet republics). Agreement and disagreement with the studies compared.

1.5       Summ up of the study, study strength and limitations

1.6 Clinical implication of the study results

  1. The conclusion is too narrow. Please rephrase the sentence in Lines 244-245 to make it more appropriate to the academic paper.

10. Some references are older than 20 years.

Author Response

Dear Reviewer 1,

Thank you for considering the manuscript Cytology-histology correlation in cervical cytology based on the recent American Society of Cytopathology guidelines (2017) at a National Medical Research Center for publication in Diagnostics (Special Issue Cyto-Histopathological Correlations in Pathology Diagnostics). 

We appreciate the time and effort that you dedicated to providing feedback on our manuscript and are grateful for the insightful comments on and valuable improvements to it. We have incorporated suggestions and highlighted changes within the manuscript. 

Please see below for a point-by-point response to your comments and concerns:

  • A THOROW ENGLISH EDITING IS REQUIRED THROUGH THE TEXT.
  • We tried to improve our English as much as we could
  • Title of the manuscript is too complex
  • We propose the new title:  Cervical cytology-histology correlation based on the American Society of Cytopathology guideline (2017) at Russian National Medical Research Center for Obstetrics, Gynecology and Perinatology
  • Introduction - It is good to introduce potential readers into the details of the territory and population to ensure that an international audience will understand the proportion of analyzed materials. Please provide information about the female population in the region it age-distribution. It is important because cervical cancer has a bimodal age distribution, with the majority of cases occurring among women in their 30s and 40s, the age at which women are often raising families and ensuring the financial viability of their families and communities. Which method is used in the country for Cervical cancer screening?
  • Thank you for pointing this out. We decided to revise the introduction and add the information: see P1L38-43
  • Line 37 in the introduction part – I suggest to be more careful with the definitions. Cytology smear is not = PAP test. Cervical cytology in the past was done by Romanovsky method, while PAP test abbreviation is related to the Papanicolaou method.
  • Thank you for pointing this out. We decided to choose ‘Pap-test’
  • The methods section does not include the detailed description of the statistical methods employed. Please make it more detailed as any original research theoretically should be reproducible.
  • Thank you for suggesting it. We decided to provide more information about statistics. See P3L121-133
  • Inclusion/exclusion criteria need to be specified.
  • Thank you for pointing this out. We expanded the information about the inclusion/exclusion criteria. See P2L83-86
  • The results part is described in a great details. Clear and interesting.
  • Thank you for your high mark.
  • The discussion part should be recomposed. The authors performed a very interesting study that should be discussed appropriately. Please cover the following
  • Thank you for suggesting it. We completely recomposed the discussion section due to your recommendations: See P8L207-245 and P9L246-284
  • The conclusion is too narrow. Please rephrase the sentence in Lines 244-245 to make it more appropriate to the academic paper.
  • Thank you for suggesting it. We tried to made it more detailed. See P9L286-292 and P10L293-298
  1. Some references are older than 20 years.
  • Thank you for pointing this out but we suppose them as the fundamental resources and could not delete them.

Reviewer 2 Report

The authors describe their study of cervical cyto-histological correlation using ASC guidelines 2017. There are certain issues that need to be addressed as detailed below:

  1. The article has 27% plagiarism using the plagiarism checker tool. The authors need to recheck the script and reduce the plagiarism percentage. 
  2. When the study has been conucted between 2019 and 2021, why was BTS 2001 used for cytology reporting? BTS 2014 should have been used for the same. 
  3. The color coding of Table 3 is incomplete. Please complete it according to the coding. 
  4. As per the methods, cytological ASCUS and histological HSIL was classified as major undercall. However, in results, the same has been included as minor undercall. Please relook at this contradiction. 
  5. The discrepant cases should have been reviewed by the same cytopathologists who reported the Pap smear initially. This procedure helps in training and self-learning. 
  6. "Hypodiagnosis". Is there such a word in the dictionary?
  7. "The minor and the major discrepancies of HSIL (PAP test) after double vision cytology ". What is double vision cytology?
  8. "Comparing our results with the Indian study showed less major discrepancies after double revision: 1.8% and 3.5% respectively". Were the gynecologic cytopathologists blinded to the fact that they were reviewing the discrepant cases? If not, then this introduces a bias in cytologic interpretation. What is the advantage of second review by gynecologic cytopathologists? 
  9. "Comparing our results with the Indian study showed less major discrepancies after double revision: 1.8% and 3.5% respectively, but there were differences according to minor discrepancies and agreement: 7.7% and 3.2% for minor discrepancies and 37.4% and 11.7% for agreements respectively" This statement is confusing. Please explain. 
  10. In the 273 cases included in this study, did the authors not detect any AGC cases? It is unusual to detect two AIS+ cases and no AGC. 

Author Response

Dear Reviewer 2,

Thank you for considering the manuscript Cytology-histology correlation in cervical cytology based on the recent American Society of Cytopathology guidelines (2017) at a National Medical Research Center for publication in Diagnostics (Special Issue Cyto-Histopathological Correlations in Pathology Diagnostics). 

We appreciate the time and effort that you dedicated to providing feedback on our manuscript and are grateful for the insightful comments on and valuable improvements to it. We have incorporated suggestions and highlighted changes within the manuscript. 

  1. The article has 27% plagiarism using the plagiarism checker tool. The authors need to recheck the script and reduce the plagiarism percentage.
  • We rewrote the huge part of the manuscript so the percentage of plagiarism decreased to 10%. We attached the plagiarism report to the revised version of our manuscript.
  1. When the study has been conducted between 2019 and 2021, why was BTS 2001 used for cytology reporting? BTS 2014 should have been used for the same
  • Thank you for pointing this out, we changed this issue and a corresponding citation: See P2L87
  1. The color coding of Table 3 is incomplete. Please complete it according to the coding. 
  • Thank you for pointing this out, we colored the table completely: See P5L150, P6L160 and P7L188
  1. As per the methods, cytological ASCUS and histological HSIL was classified as major undercall. However, in results, the same has been included as minor undercall. Please relook at this contradiction. 
  • Thank you for this suggestion, we checked it with the guide and corrected: See P3L105-107
  1. The discrepant cases should have been reviewed by the same cytopathologists who reported the Pap smear initially. This procedure helps in training and self-learning. 
  • While we appreciate your feedback, we respectfully disagree. It is excellent method to evaluate inter-observer reproducibility (with blinded initial report) or training not-experienced cytologists (with expert report available). However, the ASC guide required the cases with discrepancies to be reviewed by cytopathologists (being initially diagnosed by cytotechnologists). There are not any cytotechnologists in our Research Center that is why we used general cytopatholoists’ initial reports instead for the first round of cytological assessment.
  1. "Hypodiagnosis". Is there such a word in the dictionary?
  • Thank you for this correction. We changed the word to underdiagnostics: P8L222
  1. "The minor and the major discrepancies of HSIL (PAP test) after double vision cytology ". What is double vision cytology?
  • We meant “review cytology by the gynecologic cytologists” and changes this sentence: P3L111-112 and See P6L180
  1. "Comparing our results with the Indian study showed less major discrepancies after double revision: 1.8% and 3.5% respectively". Were the gynecologic cytopathologists blinded to the fact that they were reviewing the discrepant cases?
  • No, the gynecologic cytopathologists were not blinded to the fact that they were reviewing the discrepant cases but they were blinded to the diagnose. It can actually introduce a bias in cytologic interpretation, as you mentioned, but we conducted investigation strictly according to the guide so correlation the results between investigations planning with this guide can level out this bias because this is the common point for all such investigations.
    • What is the advantage of second review by gynecologic cytopathologists? 
  • Gynecologic cytopathologists are more experienced in the cervical pathology field and the differences in preparation between cytotechnologists and cytologists in abroad investigations can be compared with differences in preparation between general and gynecologic cytologists in Russia
  1. "Comparing our results with the Indian study showed less major discrepancies after double revision: 1.8% and 3.5% respectively, but there were differences according to minor discrepancies and agreement: 7.7% and 3.2% for minor discrepancies and 37.4% and 11.7% for agreements respectively" This statement is confusing. Please explain.
  • Thank you for this suggestion. We completely changed this paragraph with more clear data to compare: See P8L234-238
  1. In the 273 cases included in this study, did the authors not detect any AGC cases? It is unusual to detect two AIS+ cases and no AGC.
  • We included our AGC cases in the “LSIL, AGC, AGC-ECX, AGC-EMC” group according to ASC guide.

Reviewer 3 Report

I have some important questions about the study, especially its design,
which was poorly defined

Methods

The authors said in line 61: “This was a retrospective study”. Is it a retrospective cohort or a crosse sectional using files data?

 On the other hand, in line 87, the authors said: “For positive tests with acetic acid (VIA), we pro- 87 vided concurrent cervical cytology and colposcopy-directed biopsy to minimise loss of 88 follow-up for these patients”.  I think this is a little confusing. How could it be retrospective?

Line 90 ; “PAP-test was performed before 90 colposcopy if not done earlier.”  Could the sampling before colposcopy cause some limitation to colposcopy?

Results:

Line 122: “The mean period between the cervical 122 cytology and corresponding histology was 23 days (range 0-105 days).”. Wouldn't that be 105 days too long?

Discussion

Line 192: “first large-scale data analysis”. Can we consider 327 a large-scale study for the purpose of this study?

Author Response

Dear Reviewer 3,

Thank you for considering the manuscript Cytology-histology correlation in cervical cytology based on the recent American Society of Cytopathology guidelines (2017) at a National Medical Research Center for publication in Diagnostics (Special Issue Cyto-Histopathological Correlations in Pathology Diagnostics). 

We appreciate the time and effort that you dedicated to providing feedback on our manuscript and are grateful for the insightful comments on and valuable improvements to it. We have incorporated suggestions and highlighted changes within the manuscript. 

  • The authors said in line 61: “This was a retrospective study”. Is it a retrospective cohort or a crossed sectional using files data? On the other hand, in line 87, the authors said: “For positive tests with acetic acid (VIA), we provided concurrent cervical cytology and colposcopy-directed biopsy to minimized loss of 88 follow-up for these patients”.  I think this is a little confusing. How could it be retrospective?
  • Thank you for pointing this out. In our investigation we included cytological smears, histological biopsies and files data from National Medical Research Center for Obstetrics, Gynecology and Perinatology (Russia). According to ASC guide, smears with discrepancies were reviewed by second cytopathologist. The patients were not called for special examination review and none of manipulations were done specially for this investigation. Consequently, this research can be called “retrospective cohort study” and we mentioned it in our manuscript.
  • Line 90; “PAP-test was performed before colposcopy if not done earlier.”  Could the sampling before colposcopy cause some limitation to colposcopy?
  • Thank you for pointing this out, we did not mention the gap between PAP-test sampling and colposcopy which usually lasted 2-3 days. But we decided to remove this sentence from the manuscript to avoid confusion.

  • Line 122: “The mean period between the cervical 122 cytology and corresponding histology was 23 days (range 0-105 days).”. Wouldn't that be 105 days too long?

  • Thank you for this suggestion, but according to ASC guide for quality assurance purposes, 100 days and up to 365 days may be more appropriate. This interval is also mentioned in the follow article: Renshaw AA, Granter SR. Appropriate follow-up interval for biopsy confirmation of squamous intraepithelial lesions diagnosed by cervical smear cytology. American Journal of Clinical Pathology. 1997;108(3):275-279.
  • Line 192: “first large-scale data analysis”. Can we consider 327 a large-scale study for the purpose of this study?
  • Thank you for pointing this out. We removed the exaggerated definition from our article.

Round 2

Reviewer 1 Report

Dear Atuhors,

Thank you very much for considering my comments. The text of the manuscript has been improved a lot. However, some of my comments were misinterpreted. Please provide description of the abbreviation at the point of the first mention. PAP-test is an abbreviation of Papanicolaou test, therefore, when you have mentioned it for the first time in the text, you should have provided the abbreviation details. Please do it.

The text still needs editing by using any English editing tool or with the help of a native speaker. In the current version, it is difficult to follow the meaning o the manuscript due to the language inconsistency.

Overall, this paper needs to be proofread again. There are several instances with:

  1. Missing words within sentences, so they do not flow/read appropriately
  2. Wrong tenses of words are used (past/present)
  3. Wrong grammar using singular/plural subject matter and appropriate verbs
  4. Misspellings

Author Response

Reviewer 1 (second round)

Dear Reviewer 1,

Thank you for considering the manuscript Cytology-histology correlation in cervical cytology based on the recent American Society of Cytopathology guidelines (2017) at a National Medical Research Center for publication in Diagnostics (Special Issue Cyto-Histopathological Correlations in Pathology Diagnostics). 

We appreciate the time and effort that you dedicated to providing feedback on our manuscript and are grateful for the insightful comments on and valuable improvements to it. We have incorporated suggestions and highlighted changes within the manuscript. 

Please see below for a point-by-point response to your comments and concerns:

  • PAP-test is an abbreviation of Papanicolaou test, therefore, when you have mentioned it for the first time in the text, you should have provided the abbreviation details

  • We added abbreviation details where it requires.

  • The text still needs editing by using any English editing tool or with the help of a native speaker. In the current version, it is difficult to follow the meaning o the manuscript due to the language inconsistency.

  • We used English-editing service and provide a certificate for the new version of our manuscript.

Reviewer 2 Report

The authors have addressed a few of the concerns raised earlier. However, a few points still remain:

  1. The manuscript requires major English editing. In most places, it does not have readability. Please take help from a native English user or English editing software.
  2. The authors have clubbed all AGC cases with LSIL. In contrast, the indicative discrepancy assessment grid provided in the AGC guidelines group AGC/ AGC-Endocervical/ AGC-endometrial with LSIL while AGC-neoplastic is grouped with HSIL/ ASC-H for obvious connotations. The authors need to revisit their initial cytological categorization. 
  3. The authors mention that they have used TBS 2014. However, in TBS 2014, the term 'ASCUS' is not used. It has been replaced with ASC-US to make way for ASC-H. I believe that the authors should conduct their study again with the set criteria for cytologic and histologic categorization. 
  4. Were the authors performing cervical CHC prior to the ASC 2017 guidelines? If yes, did they find any advantages of the new guidelines and the discrepancy assessment grid?
  5. 'We strictly adhered to the guideline instructions and applied CHC protocol only on discrepancy cases.' The ASC 2017 guidelines mention that correlations are to be done in all cases, either in real-time or periodic retrospective. The discrepant cases need to be reviewed. The study by Gupta et al mentioned that 100% rapid rescreening by cytopathologists after cytotechnologists' review was a routine procedure. Hence, the discrepant cases were reviewed by the same trained gynecologic cytopathologists. 
  6. How did the results of this study improve the cytologic screening of the general cytopathologists in the authors' institution?
  7. A sensitivity of 84.6% and specificity of 64% for Pap smear in cervical cancer screening is not in keeping with the latest reports. 
  8. The authors mention both LBC and conventional cytology being used. Were both techniques used in all patients or patients had one of the two procedures? If LBC was used in some patients and conventional in other, the CHC results of both are likely to differ especially with general cytopathologists who are not trained in LBC. This is a serious issue to be looked into by the authors. 

Author Response

Reviewer 2 (second round)

Dear Reviewer 2,

Thank you for considering the manuscript Cytology-histology correlation in cervical cytology based on the recent American Society of Cytopathology guidelines (2017) at a National Medical Research Center for publication in Diagnostics (Special Issue Cyto-Histopathological Correlations in Pathology Diagnostics). 

We appreciate the time and effort that you dedicated to providing feedback on our manuscript and are grateful for the insightful comments on and valuable improvements to it. We have incorporated suggestions and highlighted changes within the manuscript. 

  1. The manuscript requires major English editing. In most places, it does not have readability. Please take help from a native English user or English editing software.
  • We used English-editing service and provide a certificate for the new version of our manuscript.

  1. The authors have clubbed all AGC cases with LSIL. In contrast, the indicative discrepancy assessment grid provided in the AGC guidelines group AGC/ AGC-Endocervical/ AGC-endometrial with LSIL while AGC-neoplastic is grouped with HSIL/ ASC-H for obvious connotations. The authors need to revisit their initial cytological categorization.
  • Thank you for pointing this out. There were not any patients with AGC-favor neoplastic in our cohort, so we used the name “AGC” only for patients with AGC-NOS (endocervical). We added the clarified information to the manuscript about this detail (recall the group “AGC-NOS”). In addition, we provided the information about patients with HSIL, ASC-H and AGC separately in the Table 1 not to be confused.\

  1. The authors mention that they have used TBS 2014. However, in TBS 2014, the term 'ASCUS' is not used. It has been replaced with ASC-US to make way for ASC-H. I believe that the authors should conduct their study again with the set criteria for cytologic and histologic categorization. 
  • Thank you for this suggestion. Although we did have used TBS 2014 with terms ASC-US and ASC-H but somewhere we missed hyphen in the abbreviation “ASC-US”. We corrected these mistakes.

  1. Were the authors performing cervical CHC prior to the ASC 2017 guidelines? If yes, did they find any advantages of the new guidelines and the discrepancy assessment grid?
  • Thank you for pointing this out. No, we did not perform CHC previously.

  1. 'We strictly adhered to the guideline instructions and applied CHC protocol only on discrepancy cases.' The ASC 2017 guidelines mention that correlations are to be done in all cases, either in real-time or periodic retrospective. The discrepant cases need to be reviewed. The study by Gupta et al mentioned that 100% rapid rescreening by cytopathologists after cytotechnologists' review was a routine procedure. Hence, the discrepant cases were reviewed by the same trained gynecologic cytopathologists. 
  • Thank you for pointing this out. A we have already mentioned previously, there are not any cytotechnologists in our Research Center that is why we used general cytopatholoists’ initial reports for the first round of cytological assessment. In the ASC guide there is a recommendation for discrepancies to be to “reviewed by the lead cytotechnologist with the cytotechnologist who initially signed out the case and assessed as a screening or sampling error “. We kept this recommendation and the more experienced in gynecologic cytology cytopathologist (the analog of lead cytotechnologist) reviewed discrepancies after (and with) general cytopathologist (the analog of the cytotechnologist who initially signed out the case). We added the information about collaboration of general cytopathologist and gynecological cytopathologist for teaching purpose.

  1. How did the results of this study improve the cytologic screening of the general cytopathologists in the authors' institution?
  • We could mention that all discrepancies were analyzed by general cytopathologist who initially signed out the case and his/her mentor (gynecologic cytopathologist who made corrections). Whole slide images of discrepant cases were included into the training sets for other general cytopathologists learning.

  1. A sensitivity of 84.6% and specificity of 64% for Pap smear in cervical cancer screening is not in keeping with the latest reports

  • While we appreciate your feedback, we respectfully disagree. A meta-analysis performed in 2017 demonstrated that cytology sensitivity vary from 76.2% to 87.6 depending from the category (ASCUS, LSIL, HSIL) [1]. It is very close to our results. Last year published results based on Obstetrics & Gynecology Hospital of Fudan University data demonstrated 67-68% sensitivity and 68-75% specificity[2] which is even lower than our parameters. Moreover, the specificity and sensitivity of the cytology method still vary a lot because of many confounding factors (LBC versus conventional cytology, experience of the cytologists etc.

  1. The authors mention both LBC and conventional cytology being used. Were both techniques used in all patients or patients had one of the two procedures? If LBC was used in some patients and conventional in other, the CHC results of both are likely to differ especially with general cytopathologists who are not trained in LBC. This is a serious issue to be looked into by the authors. 
  • We do not perform both conventional smears and LBC smears in all cases so some patients have the conventional smears done and some – LBC. Although general cytologists trained to diagnose as conventional as LBC smears for at least 5 years so it is not impactful source of screening discrepancies from our point of view.

[1] A. Castanon, R. Landy, D. Michalopoulos, R. Bhudia, H. Leaver, Y.L. Qiao, F. Zhao and P. Sasieni. Journal of Global Oncology 2017 3:5, 524-538

[2]J. Jun and Y. Chao-Yan. "Analysis of the efficacy of liquid-based cytology combined with HPV genotypes in screening cervical lesions in women of different ages" Journal of Laboratory Medicine, vol. 44, no. 3, 2020, pp. 151-156

Reviewer 3 Report

I'm still finding the study design confusing. It doesn't seem longitudinal to me, where each case was followed in the timeline.

Author Response

Reviewer 3 (second round)

Dear Reviewer 3,

Thank you for considering the manuscript Cytology-histology correlation in cervical cytology based on the recent American Society of Cytopathology guidelines (2017) at a National Medical Research Center for publication in Diagnostics (Special Issue Cyto-Histopathological Correlations in Pathology Diagnostics). 

We appreciate the time and effort that you dedicated to providing feedback on our manuscript and are grateful for the insightful comments on and valuable improvements to it. We have incorporated suggestions and highlighted changes within the manuscript. 

  • I'm still finding the study design confusing. It doesn't seem longitudinal to me, where each case was followed in the timeline.
  • Thank you for this suggestion. We also do not suppose that our study is a longitudinal one as a longitudinal study is a type of correlational research study that involves looking at variables over an extended period of time. Actually our study was retrospective and it would be correct to name it retrospective cross-sectional descriptive study (as we changed in the manuscript)

Round 3

Reviewer 2 Report

The authors have not addressed some of the important concerns raised earlier. A few examples are cited below:

  1. Though the TBS system in text has been changed to 2014, the reference is still of 2001 classification. This goes to show that the authors did not actually use the 2014 system.
  2. It was highlighted earlier that a cytologic report of ASC-US with histology of HSIL is a major undercall. However, the authors have still included the same as minor undercall. Have the authors actually corrected their manuscript according to the earlier comments? I doubt that. 
  3. The cases where agreement was reached by gynecologic cytopathologists should have been described in detail, highlighting the areas of learning for the general cytopathologists/ cytotechnologists. 
  4. English language editing is still required at many places. 
  5. What was the gold standard (general cytopathologists or gynecologic cytopathologists) for calculation of sensitivity/ specificity/ PPV?

Author Response

Reviewer 2 (third round)

Dear Reviewer 2,

Thank you for considering the manuscript Cytology-histology correlation in cervical cytology based on the recent American Society of Cytopathology guidelines (2017) at a National Medical Research Center for publication in Diagnostics (Special Issue Cyto-Histopathological Correlations in Pathology Diagnostics). 

We appreciate the time and effort that you dedicated to providing feedback on our manuscript and are grateful for the insightful comments on and valuable improvements to it. We have incorporated suggestions and highlighted changes within the manuscript. 

  1. Though the TBS system in text has been changed to 2014, the reference is still of 2001 classification. This goes to show that the authors did not actually use the 2014 system.
  • Thank you for pointing this out: we checked our manuscript and revealed that the reference was actually still of 2001 due to technical error. We replaced this reference with Bethesda 2014 and deleted Bethesda 2001 reference.
  1. It was highlighted earlier that a cytologic report of ASC-US with histology of HSIL is a major undercall. However, the authors have still included the same as minor undercall. Have the authors actually corrected their manuscript according to the earlier comments? I doubt that.
  • Thank you for pointing this out. We checked this point in material and methods section and noticed that we worded this sentence poorly. We corrected this paragraph in the manuscript:

Thus, according to ASC guideline, we can highlight two cases with major undercall:

1) Negative cytology result for intraepithelial lesions or malignancy (NILM) while a biopsy shows high-grade squamous intraepithelial lesion (HSIL) or an adenocarcinoma.
2) Cytological result ASC-US/AGC-NOS while a biopsy shows  squamous CA.

  1. The cases where agreement was reached by gynecologic cytopathologists should have been described in detail, highlighting the areas of learning for the general cytopathologists/ cytotechnologists.
  • Thank you for pointing this out. We added the detailed analysis if the interpretive errors to the manuscript.
  1. English language editing is still required at many places.
  • In addition to MDPI English editing proved with the certificate (attached as an additional document), the native speaker Adam Hunt has also checked the manuscript.

  1. What was the gold standard (general cytopathologists or gynecologic cytopathologists) for calculation of sensitivity/ specificity/ PPV?
  • Thank you for pointing this out. The gold standard for calculation of sensitivity/ specificity/ PPV was gynecologic cytopathologists.